# Impact of Relative Biologic Effectiveness for Proton Therapy for Head and Neck and Skull-Base Tumors: A Technical and Clinical Review

**DOI:** 10.3390/cancers16111947

**Published:** 2024-05-21

**Authors:** Adam L. Holtzman, Homan Mohammadi, Keith M. Furutani, Daniel M. Koffler, Lisa A. McGee, Scott C. Lester, Mauricio E. Gamez, David M. Routman, Chris J. Beltran, Xiaoying Liang

**Affiliations:** 1Department of Radiation Oncology, Mayo Clinic, Jacksonville, FL 32224, USA; 2Department of Radiation Oncology, Mayo Clinic, Phoenix, AZ 85054, USA; 3Department of Radiation Oncology, Mayo Clinic, Rochester, MN 55905, USA

**Keywords:** head and neck and skull-base cancers, proton therapy, relative biological effectiveness

## Abstract

**Simple Summary:**

Proton therapy is a crucial tool for head and neck and skull-base cancers, offering benefits over photon therapy by lowering the risk of adverse effects. However, its full potential could be further explored by better characterizing the uncertainties related to its relative biological effectiveness. Addressing these uncertainties is crucial for maximizing the potential of proton therapy. We explore the significance of proton therapy’s biological impact in these cancers, review relative biological effectiveness uncertainties and modeling, and examine clinical outcomes and evidence linking specific biological factors to patient adverse effects. Additionally, we review the current clinical practices and provide insights into innovative developments and their future clinical implementation.

**Abstract:**

Proton therapy has emerged as a crucial tool in the treatment of head and neck and skull-base cancers, offering advantages over photon therapy in terms of decreasing integral dose and reducing acute and late toxicities, such as dysgeusia, feeding tube dependence, xerostomia, secondary malignancies, and neurocognitive dysfunction. Despite its benefits in dose distribution and biological effectiveness, the application of proton therapy is challenged by uncertainties in its relative biological effectiveness (RBE). Overcoming the challenges related to RBE is key to fully realizing proton therapy’s potential, which extends beyond its physical dosimetric properties when compared with photon-based therapies. In this paper, we discuss the clinical significance of RBE within treatment volumes and adjacent serial organs at risk in the management of head and neck and skull-base tumors. We review proton RBE uncertainties and its modeling and explore clinical outcomes. Additionally, we highlight technological advancements and innovations in plan optimization and treatment delivery, including linear energy transfer/RBE optimizations and the development of spot-scanning proton arc therapy. These advancements show promise in harnessing the full capabilities of proton therapy from an academic standpoint, further technological innovations and clinical outcome studies, however, are needed for their integration into routine clinical practice.

## 1. Introduction

Many studies support the role of proton therapy in head and neck cancer to reduce integral dose, decreasing the incidence of acute and late toxicities related to the low and moderate dose bath. This manifests clinically as lower incidences of dysgeusia, feeding tube dependence, xerostomia, and secondary malignancies, and reduced neurocognitive dysfunction [1,2,3,4,5,6]. Clinical outcomes studies from those treated at MD Anderson Cancer Center have shown that patients with oropharyngeal cancer treated with proton therapy have low rates of acute feeding tube dependence and favorable rates of post-treatment xerostomia, among other adverse effects [1,2]. Furthermore, a recent study from Memorial Sloan Kettering Cancer Center compared intensity-modulated proton therapy (IMPT) to intensity-modulated radiation therapy for oropharyngeal cancer and demonstrated significantly reduced acute toxicity burden with IMPT, with few chronic toxic effects and favorable oncologic outcomes [6].

Moving beyond the reduction in integral dose, an emerging area of clinical research and the main focus of this paper is the enhanced biologic effects of proton therapy secondary to increased relative biologic effectiveness (RBE) compared with photon therapy. Clinical studies have shown data that indicates promising outcomes with particle therapy, which in turn suggests that, in addition to reducing integral dose, protons may have different biologic effects. These effects manifest both within the target volume, including potential cell kill and treatment efficacy, and throughout the surrounding organs at risk (OARs). Herein, we discuss the clinical importance of RBE within the proton therapy treatment volume and in adjacent serial OARs of the head and neck and skull-base and review the uncertainties and the modeling of proton RBE. We explore clinical outcomes data and clinical evidence regarding the correlation between enhanced linear energy transfer (LET) and RBE with patient toxicity in head and neck and skull-base cancer. Furthermore, we review current clinical practices and provide insight into innovative developments. We discuss the challenges faced in implementing these innovations and offer outlooks on how these advancements may shape future medical practices.

## 2. Clinical Outcomes Data

Regarding the potential impact of treatment efficacy related to RBE, studies have shown promising tumor control outcomes with proton therapy compared with photon therapy. In the largest sinonasal cancer treatment study to date, Patel et al. [7] performed a systematic review and meta-analysis demonstrating a disease-free survival and overall survival benefit for proton beam therapy compared with photon modalities for sinonasal malignancies. From a biologic standpoint, in vitro head and neck cancer cell lines show that samples positioned within the middle of the spread-out Bragg peak may exhibit an RBE higher than the current clinical practice standard of a constant RBE of 1.1 [8]. Additionally, evidence suggests variable RBE with protons, enhanced at the distal Bragg peak, with several studies on OARs indicating that the enhanced RBE poses a challenge for OARs at the distal end [9,10,11,12]. Another consideration not related to RBE, however, may include the way in which the reduced dose to surrounding normal OARs with proton therapy allows for improved target coverage, particularly when treating malignancies near critical neurosensitive organs such as those with neurotropic spread or that involve perisellar or skull-base regions. 

Head and neck normal tissue anatomy has a wide range of defined OARs and wide heterogeneity in alpha/beta ratios, serial and parallel structures because of the multitude of primary site locations, local and regional patterns of failure and the potential for neurotropic spread [13]. Central nervous system radionecrosis for the treatment of head and neck and skull-base tumors ranges from 1% to 5% [14]. However, several groups have shown that radiographic evidence of parenchymal central nervous system imaging changes are higher with particle therapy [14]. Song et al. [15] reviewed treatment of 77 patients with meningioma; 38 received proton therapy and 39 received photon therapy. While grade 2 or higher adverse effects were 10% or less in both cohorts, the 2-year cumulative incidences of magnetic resonance imaging changes were 26.8% after proton therapy and 5.3% after photon therapy (*p* = 0.02). For tumors of the central skull base, the most relevant toxicity was the manifestation of temporal radionecrosis [15]. Zhang et al. [16] reviewed 566 patients treated with either double-scattered protons or intensity-modulated radiation therapy. They showed that tolerance curves of proton therapy were lower than that of photon treatments at all dose levels. In fact, the dose tolerance at D1% was 10 Gy less for protons compared with photons, and the RBE for temporal lobe enhancement from proton treatments were calculated to be 1.18 [16]. This is similar to a University of Washington study that showed that, while there were statistically different changes in radiographic brain images, the overall rates of toxicity were low [15]. 

Several models have been developed to predict temporal radionecrosis following proton therapy. In 2015, McDonald et al. [17] published the outcomes of 66 patients treated with uniform scanning 3-dimensional conformal proton therapy to a median dose of 75.6 GyRBE. With a median follow-up of 31 months, the 3-year estimate of any grade temporal lobe radiation necrosis was 12.4%. The authors showed that only dose volume relationship was associated with risk of radiation necrosis at all dose levels up to 70 GyRBE. They modeled a 15% 3-year risk of any-grade temporal lobe radiation necrosis when the absolute volume (V) of a temporal lobe receiving 60 GyRBE exceeded 5.5 cm^3^, or V70 GyRBE greater than 1.7 cm^3^ [17]. This differs slightly from a publication by Schroder et al. in 2022 [18] in which the authors evaluated nearly 300 patients with a median radiologic follow-up of over 4.5 years. Their results show that, not only clinical dose, but also age, V40 GyRBE, hypertension, and dose to at least 1 cc (D1cc) in the temporal lobe, are associated with grade 2 necrosis or greater. This was cross-valuated and used to develop an age*prescription-dose*D1cc (Gy)*hypertension patient-wise model with maximum area under the receiver operating characteristic curve of 0.76 [18].

One concern with proton therapy is distal end ranging of high RBE protons into the brainstem. Clinical data and outcomes data suggest that, despite the potential risk of higher brainstem toxicity related to this phenomenon, there are low rates of injury following proton therapy [19]. Long-term outcomes reported by Weber et al. [19] from the Paul Scherrer Institute, following treatment of 222 patients with chordoma and chondrosarcoma found that, despite nearly one-third of patients having residual tumor abutting the brainstem, no patient experienced brainstem toxicity. A group from the University of Florida conducted investigations in both adult [20] and pediatric skull-base cancer [21]. Within the adult cohort, Holtzman et al. [20] have reported that the 5-year cumulative incidence of grade 2 or higher brainstem injury was 1.3% (95% CI, 0.25–4.3%). Indelicato et al. [21] reported similar results in a pediatric population, showing that the 2-year cumulative incidence of overall brainstem toxicity was 3.8% (SD 1.1%), and incidence of grade 3 or higher toxicity was 2.1% (SD 0.9%). A National Cancer Institute consortium report [22] showed that, after adopting modified radiation guidelines, the actuarial rate of grade 2 or higher brainstem toxicity was successfully reduced from 12.7% to 0% at 1 center, and there were no differences between the risk of radionecrosis between proton and photon therapy. While these cohorts did not use LET optimization, the consortium did note that increased capability exists to incorporate LET optimization [22]. However, further research is warranted to thoroughly investigate the capabilities of LET- and RBE-guided planning. 

Regarding the optic apparatus, proton therapy has been used with acceptable rates of toxicity. Potential toxicity when treating the head and neck may include damage to any structure from the anterior orbit to posteriorly along nervous tissue. Holliay et al. reported on 20 patients treated at MD Anderson Cancer Center who underwent an orbital-sparing surgery followed by proton therapy for ocular tumors [23]. They noted 3 developed chronic grade 3 epiphora and 3 developed grade 3 exposure keratopathy; 4 patients experienced a decrease in visual acuity from baseline [23]. Kountouri et al. [24] evaluated patients with skull-base cancer treated with pencil beam scanning for radiation-induced optic neuropathy (RION). They found that, in univariate analyses, age (<70 vs. ≥70 years; *p* < 0.0001), hypertension (*p* = 0.0007), and tumor abutting the optic apparatus (*p* = 0.012) were associated with RION [24]. Similarly, Li et al. [25] showed that 17 of 514 patients developed RION following proton-based therapy at the Massachusetts General Hospital. While the incidence of RION was 1% among patients receiving less than 59 GyRBE to the optic pathway, it was 5.8% among those receiving 60 GyRBE or greater [25]. These results are consistent with those reported by De Leo et al., where the 5-year incidence of vision loss was 2.1% (95% CI, 0.9–4.9%) [26]. While no adverse events were observed with a maximum dose less than 60 GyRBE delivered to the anterior optic pathway, the crude rate was 3.6% for doses of 60 GyRBE or greater, with all events occurring between 60 and 65 GyRBE [26].

Regarding the oral cavity and mucosa, study results show notable differences in RBE between proton and photon therapy. Singh et al. [27] reviewed 122 radiation therapy–naive patients with oral and oropharyngeal cancer treated with proton radiotherapy at Memorial Sloan Kettering Cancer Center. The 5-year rate of osteoradionecrosis was 11.5% with the posterior ipsilateral mandible within the radiation field being the site most involved [27]. These results are also seen in patients treated at Mayo Clinic [28]. Their findings suggest that RBE is larger than 1.1 at moderate doses (40–60 GyRBE) with high LET for mandible osteoradionecrosis, and that RBEs are underestimated in current clinical practice [28]. Follow-up studies evaluating posttreatment positron emission tomography–computed tomography suggest that mucosal RBE may also show clinical differences. Gelover-Reyes et al. [29] reviewed 19 patients without local failure treated with 50 GyRBE or greater to oropharyngeal mucosa in the adjuvant setting following primary surgery. They note that these mucosal areas showed increased post-therapy maximum standardized uptake values on follow-up positron emission tomography–computed tomography compared with patients receiving photon therapy, with the area of the clinical target volume at the primary site. Positron emission tomography avidity on these scans was postulated to represent clinical mucosal injury and inflammatory changes secondary to dose escalation through RBE [29].

In summary, proton therapy has potential advantages in regard to enhanced biologic effects (Table 1). While differences in normal tissue RBE have been identified within high-dose regions, there does not appear to be a clinically meaningful increase in toxicities rates. Moreover, there are two ongoing phase III proton and photon modality comparison studies that are ongoing. One of which is a multi-institutional phase II/III being led by Frank et al. [30], and the other by Thompson et al., the TORPEdO study of the National Health Service in the United Kingdom [31]. 

## 3. Proton RBE: Uncertainties and Modeling

RBE of protons is defined as the ratio of the absorbed doses needed to produce an equivalent biological response between proton beam irradiation and a reference radiation, as follows [32]: $$RBE(EndpointX)=\frac{{Dose}_{reference}(EndpointX)}{{Dose}_{proton}(EndpointX)}$$

In clinical practice, a constant RBE of 1.1 is assumed for protons, based on early in vitro and in vivo radiobiological experiments. This number is considered a conservative estimate to ensure adequate tumor coverage [32]. However, published in vitro studies have shown a significant variation in RBE among different cell lines [33]. Additionally, in vitro studies have demonstrated that RBE varies with factors such as LET, α/β ratio, biological endpoint, and dose [34,35,36,37,38,39,40], and that the relationship between cell kill, dose and LET is intricate [41]. The American Association of Physicists in Medicine Task Group report 256 [32] has provided a comprehensive review of the current knowledge of RBE.

As protons slow down at the distal end, this leads to higher LET and RBE values. Specifically, LET describes the average energy transfer per length of radiation track [42]. Protons retain most of their kinetic energy until near the end of their path, at which point they undergo rapid terminal deceleration, leading to a sharp increase in LET in the distal span [43,44,45]. This effect also extends the biological range of protons a few millimeters beyond their physical range [33,46]. The increased LET and RBE at the end of the range could potentially affect normal tissue located immediately beyond the target. Understanding the uncertainty in RBE has become an active area of research.

Numerous RBE models have been developed. These models can be classified into two main types: phenomenological and mechanistic models. Mechanism models, such as the kinetic repair–misrepair–fixation model [47], microdosimetric–kinetic model (MKM) [48], local effect model [49,50], and MCF MKM [51,52], are based on a mechanistic approach. This involves studying the physical interactions of radiation with biological molecules and cells, as well as the subsequent biological responses, such as cell survival, and then incorporating this understanding into mathematical models. In contrast, phenomenological models describe the relationship between radiation dose and biological effect by fitting a radiobiological model to empirical data, without delving into detailed biological mechanisms. Various phenomenological models have been proposed, with most based on the linear–quadratic model [53,54,55,56,57,58]. Additionally, simple linear–fit models have been developed to estimate RBE as a linear function of LET by fitting clonogenic cell survival data [59,60]. Most RBE studies in proton therapy have employed phenomenological models due to their simplicity and practicality. Rørvik et al. [61] and McNamara et al. [62] conducted comprehensive reviews on various phenomenological RBE models. Both of these reviews demonstrate that, while there is a general agreement in overall trends, significant variations among different model estimations can be observed (Figure 1). 

McMahon [63] investigated the underlying trends in different RBE models. This study compared the predictions from various RBE models and investigated the relationship of the predictions with relevant biological and physical parameters. It aimed to identify areas of conceptual agreement or disagreement among these models. The study concluded that the main variations between models stem from their treatment of biological parameters [63]. By contrast, all models consistently demonstrate strong correlations with LET. This suggests that LET, a physics property that can be accurately calculated, is a possible solution to be used as a surrogate for RBE in plan optimizations.

## 4. Clinical Evidence of Correlation between Enhanced LET/RBE and Patient Toxicity

Recently, several clinical studies have investigated the correlation between enhanced LET/RBE and patient toxicity or radiographic image changes following proton therapy. In these studies, voxel-by-voxel LET/RBE data were retrospectively generated using a variable RBE model. Subsequently, correlation studies were conducted to determine if patient toxicity or radiographic image changes are correlated with LET/RBE. Underwood et al. [64] performed a systematic review of 22 clinical studies on variable RBE. Conclusions regarding clinical evidence for variable RBE were mixed: 12 studies yielded inconclusive results, 6 reported finding clinical evidence for variable proton RBE, and 4 concluded the opposite. It is worth noting that most studies included in the review involved fewer than 20 patients. The authors acknowledge that the limited clinical evidence for variable proton RBE was likely a result of inadequate large-scale prospective data sets and challenges in retrospective studies [64].

In the context of head and neck and skull-base tumor treatment with proton therapy, Fossum et al. [65], in a study of 11 patients who underwent head and neck proton therapy, recalculated clinical treatment plans using an in-house graphics processing unit-based Monte Carlo algorithm, and a linear model was used to convert physical dose and LET into biologic dose. Both physician- and patient-reported toxicities were included. The study found a strong correlation between increased LET/RBE and toxicity in the oral cavity and oropharynx; however, this correlation was less pronounced in areas such as the brain and mandible. Importantly, higher LET and RBE did not consistently lead to adverse toxicities. The authors proposed that dose adjustments based on LET and RBE predictions hold clinical significance [65]. In another study, Niemierko et al. [66] assessed the correlation between radiographic imaging changes defined as necrosis with regions of elevated LET following proton therapy. In this retrospective review of 50 patients with head and neck, skull-base, or intracranial tumors, LET was not found to correlate with risk of brain necrosis. However, the authors acknowledge that such analysis is difficult due to the uncertainties involved in necrosis evaluation. Therefore, the RBE effects might be obscured by other confounding factors [66]. 

Wagenaar et al. [67] computed the voxel-wise product of physical dose and dose-averaged LET (LETd) (D⋅LETd) for 100 patients undergoing head and neck proton therapy. Their goal was to evaluate the feasibility of associating mean D⋅LETd parameters with patient toxicity. They concluded that a sample size of over 15,000 patients would be required to create a normal tissue complication probability model with a power of at least 80% to show the independent effect of mean D⋅LETd, which shows it is not feasible to direct a normal tissue complication probability model with D⋅LETd [67].

## 5. Current Clinical Practice and Outlook for Future Clinical Practices

The treatment planning process for head and neck and skull-base tumors can be challenging due to the complexity of the target volumes and surrounding OARs. Multifield optimization is typically used in proton treatment planning for these disease sites [68]. Multifield optimization optimizes spots from all proton fields together, using the flexibility of spot-scanning techniques to create highly modulated dose distributions. This approach can result in highly inhomogeneous doses from each individual field [69]. To account for setup and range uncertainties, robust optimization is recommended [70,71]. 

Due to concern for the sensitivity of proton therapy to changes in anatomic volume, many institutions have implemented routine on-treatment surveillance with verification imaging and robust planning techniques. Mayo Clinic found that weekly verification computed tomography quality assurance scans frequently influenced clinical decisions to replan [72]. Nearly 50% of all patients with head and neck cancer required replanning by week 5 of therapy as did over 70% of those with sinonasal cancer [72]. 

Although a constant RBE of 1.1 is adopted in current clinical practice, it is commonly agreed that ignoring the effects of variable RBE and high LET distribution may lead to unexpected adverse events. Therefore, in clinical practice, RBE uncertainties are often indirectly considered by carefully selecting beam angles to avoid directly targeting OARs [37,73]. A survey study conducted by Heuchel et al. [74] among European proton therapy centers showed that variable RBE is frequently taken into account by avoiding beams from stopping inside or in front of an OAR and by increasing the number of fields used to reduce the relative weight of critical incident angles.

### 5.1. LET/RBE-Guided Plan Optimization

In addition to the indirect approaches commonly applied in current clinical practice, various LET/RBE-guided plan optimization approaches have been proposed [75,76,77,78,79,80,81,82,83,84,85]. These approaches aim to achieve more favorable LET and RBE distributions, thereby minimizing the biological dose in OAR without compromising target volume coverage. Several methods have been introduced, including optimizing LET [75,77,79,80], variable RBE-weighted dose [82], track-end distributions [84], and introducing “clean dose” (dose deposited by low-LET protons) and “dirty dose” (dose deposited by high-LET protons), penalizing the latter in OAR [83,85]. Early work by Grassberger et al. [78] has demonstrated that pencil beam scanning provides the opportunity to optimize LET distribution without significantly altering the distribution of physical dose. Additionally, LET-guided robust optimization has been proposed to ensure both dose and LET are robust in the presence of uncertainties [75,79,86]. Therefore, LET-guided optimization offers the possibility of avoiding high LET spots within or near OARs to minimize the potential incidence of adverse events [80]. The efforts to optimize LET to a position within the tumor may also enhance the therapeutic ratio. 

In a study by Liu et al. [79], 14 head and neck tumor treatment plans were retrospectively generated using LET-guided robust optimization. The study found that high LET was redistributed from nearby OARs to tumors, albeit with a slight increase in physical dose and a minor impact on plan robustness. The LET-guided optimized plan successfully reduced the high LET volume in the brainstem, redistributed it to the target volume, and filled the cold LET area inside the target. The final dose distribution remained nearly the same as in the standard optimized plan [79]. In a study by Wan Chan Tseung et al. [81], biological dose was directly incorporated into the optimization, and inverse biological planning was applied to four patients with head and neck cancer. Results show that, compared with standard IMPT, the biologically optimal plans were able to achieve a biological dose escalation of approximately two times the physical dose for small tumor targets. Additionally, dose sparing to critical structures was improved when compared with standard IMPT [81].

Although there is ongoing research into LET/RBE-guided optimization and promising results have been presented, its implementation remains largely limited to in-house developed treatment planning systems (TPSs) or the research versions of specific TPSs. Before LET/RBE-guided optimization can be routinely implemented in clinical practice on a widespread scale, it must achieve a level of maturity that ensures user-friendliness and reproducibility. This would necessitate collaborative efforts involving research, clinical, and industry stakeholders. In a survey study on future improvements regarding RBE, the most frequent and urgent request was for more clinical evidence of possible effects due to increased RBE [74]. 

### 5.2. Spot-Scanning Proton Arc Beam Delivery

In recent years, there has been increasing interest in exploring spot-scanning proton arc (SPArc) in proton therapy. SPArc uses a dynamic rotational gantry to deliver proton beams in an arc trajectory [87,88]. While traditional particle therapy minimizes integral dose and irradiated volume by using single or few beams, SPArc delivers proton radiation from numerous gantry angles. The additional gantry angles provide more flexibility in plan optimization. Studies suggest that, compared with traditional methods, such as IMPT, SPArc plans can reduce the dose to nearby organs at risk while maintaining robust target coverage [76,77,78,79,80,81]. In addition to providing superior dosimetric distribution, SPArc’s additional beam angles facilitate effective optimization of LET. A feasibility study investigating the integration of LET into SPArc optimization and its potential clinical benefits found that with similar dose distributions, the optimization of LET distributions becomes more effective with an increasing number of beams [89]. For the brain case in this study, average LETd values for the SPArc plan exhibited a 29% increase for clinical tumor volume and decreases of 22%, 30%, 28%, and 17% for the brainstem, chiasm, and left and right optical nerves, respectively, when compared with the 3-beam standard non-LETd optimized plan. In contrast, the respective changes for the 3-beam LETd-optimized IMPT plan were only 4%, 12%, 22%, 21%, and −3% [89]. Beyond proton therapy, interest in exploring arc therapy has expanded to include heavy ion therapy [90,91]. The study involving skull-based chordoma showed that the target LETd were boosted from approximately 60 to 90 KeV/μm by carbon arc therapy [91]. The authors suggest that the favorable LET observed in heavy ion arc therapy may help mitigate the loss of treatment efficacy induced by tumor hypoxia. 

However, a SPArc study focusing on four brain tumors has indicated that, while an increased number of beams could enhance dose gradients to better spare cognitive brain structures, it might also lead to increased low-dose/low-LET volume [92]. The authors caution about the potential implications for secondary cancer risk. Conversely, another study examining five pediatric brain tumors evaluated radiation-induced second primary cancer risks [93]. Using two brain-specific second primary cancer models, the study concluded that the estimated second primary cancer risks were consistently lower for SPArc compared with volumetric modulated arc therapy and IMPT [93]. 

While SPArc shows promise, further studies are needed to fully understand the balance between improving LET distribution and the potential clinical consequences of an increase in low-dose volume. Moreover, SPArc significantly increases the complexity of treatment delivery, presenting technical challenges that necessitate further technological development and investigation. For example, SPArc requires high stability in beam current and high precision of synchronization of the beam delivery with gantry rotation. A fast energy-switching system is also essential for effective utilization of SPArc. Efforts to advance the technology for potential future clinical applications are ongoing.

## 6. Conclusions

Proton therapy offers distinct advantages in both integral dose distribution compared with photon therapy and in RBE, which can be leveraged for treatment optimization to improve the therapeutic ratio. However, the uncertainties associated with proton RBE pose challenges in current practice. Variable RBE is not yet routinely incorporated into proton therapy treatment planning, which could limit the full potential of proton therapy. Technological advancements and innovations in plan optimization and treatment delivery, such as LET/RBE optimizations and SPArc delivery, show promise, but further investigations, developments, and clinical outcome studies are necessary for their routine adoption in clinical practice.

## Figures and Tables

**Figure 1 cancers-16-01947-f001:**
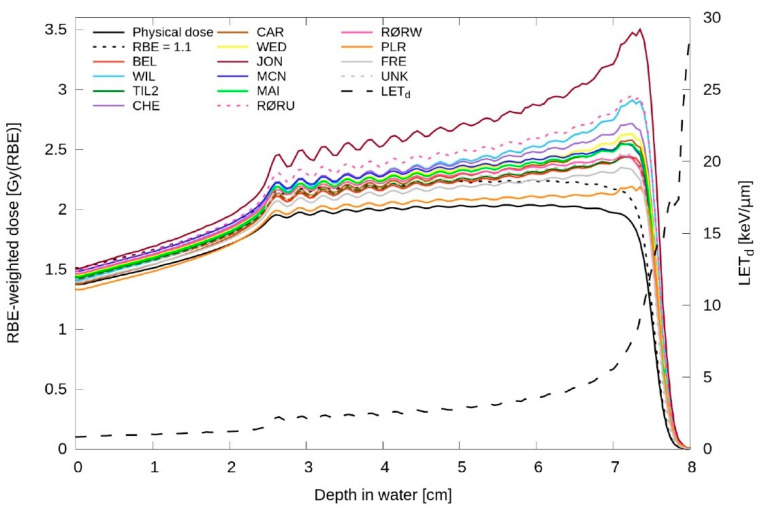
Biological dose estimations from various phenomenological RBE models from Rørvik et al. [61], Phys Med Biol 63 (18), 185,013 (2018). © Institute A physics and engineering in medicine, reproduced with the permission of IOP Publishing Ltd. BEL, CAR, CHE, FRE, JON, MAI, MCN, PLR, RØRU, RØRW, TIL2, UNK, WED, and WIL represent various phenomenological models included in the study. Please refer to Table 1 in [61] for details on each model. LETd, the dose-averaged linear energy transfer, is represented on the right axis. RBE indicates relative biological effectiveness.

**Table 1 cancers-16-01947-t001:** Summary of clinical outcomes.

Author, Year	No. of Patients	Median RT Dose	RT Modality	Findings
**Head and neck (mandible/mucosa)**				
Gelover-Reyes et al., 2023 [29]	19	≥50 GyRBE	IMPT (n = 9); VMAT (n = 10)	LET and RBE were elevated for IMPT compared with VMAT patients, documenting dose enhancement ratios of 10–30% above clinical dose
Singh et al., 2023 [27]	122			3- and 5-year rates of ORN were 5.2% and 11.5%, respectively
Yang et al., 2022 [28]	1266	≥60 GyRBE	IMPT (n = 335); VMAT (n = 931)	Mandible V40, V50, and V60 GyRBE were significantly different (*p* < 0.05) between the ORN and control group for VMAT and PBSPT Empirical RBEs of 1.58, 1.34, and 1.24, at 40, 50, and 60 Gy[RBE = 1.1], respectively
**Central nervous system (optic pathway)**				
De Leo et al., 2021 [26]	148	73.8 GyRBE	IMPT (n = 139); DS (n = 9)	No vision loss with maximum dose < 60 GyRBE delivered to the anterior optic pathway Crude rate was 3.6% with dose > 60 GyRBE
Holliday et al., 2016 [23]	20	60 GyRBE	IMPT (n = 6); DS (n = 14)	Six patients developed either chronic grade 3 epiphora or keratopathy Four patients experienced grade 2 decrease in visual acuity from baseline
Kountouri et al., 2020 [24]	216	74 GyRBE	IMPT	Grade ≥ 3 RION in 12 patients Maximum dose ≥ V60 GyRBE to optic pathway did not meet statistical significance (*p* = 0.06)
Li et al., 2019 [25]	514	75.2 GyRBE, chordoma; 70 GyRBE, chondrosarcoma	DS-based (n = 466); DS alone (n = 48)	RION was 1% among patients receiving < 59 GyRBE and 5.8% among those receiving 60 GyRBE
**Central nervous system (brainstem)**				
Indelicato et al., 2014 [21]	313	54 GyRBE	DS	Two-year cumulative incidence of any toxicity and grade ≥ 3 toxicity was 3.8% and 2.1%, respectively
Haas-Kogan et al., 2018 [22]	671	NR	Proton (n = 671); IMRT (n = 60)	Average rate of symptomatic brainstem toxicity from the three largest United States pediatric proton centers was 2.4% Brainstem injury is a rare adverse effect of radiation therapy for both photons and protons
Holtzman et al., 2022 [20]	163	73.8 GyRBE	IMPT (n = 18); DS (n = 145)	Five-year cumulative incidence of grade ≥ 2 brainstem injury was 1.3% All patients recovered baseline with medical management
**Central nervous system (parenchymal)**				
McDonald et al., 2015 [17]	66	75.6 GyRBE	DS	Three-year estimate grade ≥ 2 TRN radiation necrosis was 5.7%
Song et al., 2021 [15]	77	54 GyRBE	IMPT (n = 23); US (n = 15); IMRT (n = 39)	Two-year cumulative incidences of T1c + T2 changes were 26.8% after proton therapy and 5.3% after photon therapy (*p* = 0.02) No significant differences in symptomatic adverse events
Zhang et al., 2021 [16]	566	70 GyRBE	Proton-based (n = 60)	Dose tolerance (D1%) was 58.6 Gy for protons and 69.1 Gy for photons The RBE for temporal lobe enhancement from proton treatments were 1.18
Schroder et al., 2022 [18]	299	74 GyRBE, chordoma; 70 GyRBE, chondrosarcoma, head and neck	IMPT	Twenty-seven patients (9%) developed grade ≥ 2 TRN Prescription dose, age, V40 Gy (%), hypertension, and dose to at least 1 cc (D1cc) (Gy) in the temporal lobe were found to predict TRN Cross-validation showed that a model using age*prescription-dose*D1cc (Gy)*hypertension was superior in all described test statistics

Abbreviations: DS, double scattered; IMPT, intensity-modulated proton therapy; IMRT, intensity-modulated radiation therapy; LET, linear energy transfer; NR, not reported; ORN, osteoradionecrosis; PBSPT, pencil-beam-scanning proton therapy; RBE, relative biological effectiveness; RION, radiation-induced optic neuropathy; RT, radiation therapy; TRN, temporal lobe radionecrosis; US, uniform scanning; VMAT, volumetric-modulated arc therapy.

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
