# Peer review of "Impact of Relative Biologic Effectiveness for Proton Therapy for Head and Neck and Skull-Base Tumors: A Technical and Clinical Review"

_cancers, 2024, doi:10.3390/cancers16111947_

Round 1

Reviewer 1 Report

Comments and Suggestions for Authors

In this review, the authors described the impact of proton variable RBE on head&neck and skull base tumors. The review is ok to be published. I have some comments.

(1)  Line 131, it should be “Holliday”

(2)  Please explicitly describe LETd is dose averaged LET at its first appearance.

(3)  Line 182, please cite more papers related to experiments:

https://www.nature.com/articles/srep09850

https://www.sciencedirect.com/science/article/pii/S036030161400594X?via%3Dihub

https://www.nature.com/articles/s41598-020-60246-5

(4)  Line 194, in general two categories: phenomenological and mechanistic models. Please rewrite here. Please also introduce RMF model as a mechanistic model. Please also include Mayo clinic Florida’s MKM model.

Carlson, David J., et al. "Combined use of Monte Carlo DNA damage simulations and deterministic repair models to examine putative mechanisms of cell killing." Radiation research 169.4 (2008): 447-459.

https://www.mdpi.com/1422-0067/23/20/12491

Reviewer 2 Report

Comments and Suggestions for Authors

The authors present a comprehensive review of the literature concerning the role of the variable Relative Biological Effectiveness (RBE) in the treatment of head and neck tumors using proton therapy. The paper initially delves into the established clinical foundations of proton therapy for head and neck cancers, explores various RBE modeling approaches, discusses the relationship between Linear Energy Transfer (LET)/RBE and treatment toxicity, and introduces ongoing clinical investigations on LET optimization/evaluation. Furthermore, the manuscript anticipates the future integration of arc therapy within the realm of proton therapy for head and neck tumors, offering valuable insights and promising avenues for further research.

We may have a few remarks:

In the introduction (line 56), the challenge of considering the variability of Relative Biological Effectiveness (RBE) arises notably for tumors situated at the periphery of the treated tumor (barring specific tumor characteristics or highly altered fractionation schemes) and not so much on the treated volume. Regarding clinical outcomes, the observed improvement from line 70 to 73 may be attributed to enhanced coverage of target volumes facilitated, for instance, by the achievable dose gradient in proton therapy. This proposition is advanced as a rationale in Patel's study discussion and could be incorporated into your review as a potential explanation for the efficiency observed as much as your emphasis on a potential RBE > 1.1.

From lines 79 to 148, a significant portion of the research in Chapter 2 pertains to clinical data concerning organ toxicity in the central nervous system, specifically in the context of sinus and para-sinus irradiations, with limited focus on the broader spectrum of head and neck cancers. It would be pertinent to provide a concluding statement addressing the following questions: What are the critical head and neck organs at risk from a clinical perspective? What are their a/b values, and what imaging modalities can be utilized for objective évaluation of the toxicity attributed to an "higher than anticipated RBE value" to advance understanding in this area? Figure 1 from Rorvik et al. warrants explanation, particularly regarding the models depicted therein. What is the meaning of "BEL, WIL, TIL2" ? Chapter 5 provides compelling insights. The introduction could have been crafted by exploring the diverse methodologies for integrating the variable RBE, exemplified in the research conducted by Hahn et al. in Radiation Oncology (2022)  https://doi.org/10.1186/s13014-022-02143-x . The study delves into the ongoing trial propositions by notable researchers such as Professor Krause from Heidelberg and Professor Wagenaar from Groningen.

Finally, this manuscript offers significant contributions to current discourse and offers expansive viewpoints, anticipating the outcomes of NCT01893307.

Reviewer 3 Report

Comments and Suggestions for Authors

This article discusses the relative biological effects of proton therapy focusing on head and neck and skull-base tumors. The authors offered a comprehensive overview of current clinical practices, innovative developments, and future implementation strategies in proton therapy for head and neck and skull-base tumors, guiding the proton therapy society towards advancements in treatment modalities and patient care.

This review paper will serve as a valuable resource for the society by consolidating current knowledge, addressing challenges, and providing a roadmap for future research and clinical implementation in proton therapy for cancer treatment.
